# Efficient Reaction Systems for Lignocellulosic Biomass Conversion to Furan Derivatives: A Minireview

**DOI:** 10.3390/polym14173671

**Published:** 2022-09-04

**Authors:** Xiaofang Liu, Dayong Yu, Hangyu Luo, Can Li, Hu Li

**Affiliations:** 1Guizhou Provincial Key Laboratory for Rare Animal and Economic Insects of the Mountainous Region, College of Biology and Environmental Engineering, Guiyang University, Guiyang 550005, China; 2State Key Laboratory Breeding Base of Green Pesticide & Agricultural Bioengineering, Key Laboratory of Green Pesticide & Agricultural Bioengineering, Ministry of Education, State-Local Joint Laboratory for Comprehensive Utilization of Biomass, Center for Research & Development of Fine Chemicals, Guizhou University, Guiyang 550025, China

**Keywords:** lignocellulosic biomass, ionic liquid, deep eutectic solvent, biphasic system

## Abstract

Lignocellulosic biomass as abundant, renewable, and sustainable carbon feedstock is an alternative to relieve the dependence on fossil fuels and satisfy the demands of chemicals and materials. Conversions of lignocellulosic biomass to high-value-added chemicals have drawn much attention recently due to the high availability of sustainable ways. This minireview surveys the recent trends in lignocellulosic biomass conversion into furan derivatives based on the following systems: (1) ionic liquids, (2) deep eutectic solvents, and (3) biphasic systems. Moreover, the current challenges and future perspectives in the development of efficient routes for lignocellulosic biomass conversion are provided.

## 1. Introduction

The sustainable growth of fossil fuels and fossil-fuel product consumption has caused serious environmental issues, appealing to exploited renewable and eco-friendly sources as an alternative to overcome the depletion. The world energy requirement is predicted to enhance by 24% by the year 2040 based on the International Energy Agency’s (IEA) projection [1]. Therefore, the transformation of lignocellulosic biomass to various platform chemicals has been recognized as the best candidate to cater to the urgent demand due to the abundant reserves. For a long time, physical, biological, chemical, and combined approaches have been explored to convert lignocellulosic biomass into useful products, forming the roadmap of biorefinery technologies [2,3,4]. The utilization of lignocellulosic biomass is blocked by the degree of crystallization and polymerization of cellulose, existence of hemicellulose and lignin, degree of acetylation of hemicellulose, and so on. The major obstacles are mainly due to the crystalline nature of cellulose embedded in a matrix of hemicellulose and lignin, which is determined by the reaction system combined with various catalysts.

Many researchers have focused on pretreatment methods that single, two-stage or multi-stage approaches might provide better and more efficient strategies for oxygen-containing biomass, including cellulose, hemicellulose, and their monomers or derivatives conversion, during which hydrolysis, dehydration/alkylation, and hydrodeoxygenation (HDO) reaction processes have occurred. Single-step means require more energy consumption and lead to a range of environmental issues that restrict the production of biofuels. On the contrary, two- or multi-stage process tactics use mild reaction conditions that economize and elevate the productivity of the various lignocellulosic biomasses (Table 1).

Owing to the consideration of cost, a green and sustainable reaction medium is regarded as a crucial factor to be investigated. Herein, this minireview aims to concisely summarize the performance of the ionic liquids (ILs), deep eutectic solvents (DESs), and biphasic systems in the catalytic conversion of lignocellulosic biomass for the preparation of different biofuels, especially furan derivatives by cost-effective means. The lignin upgrading for significant platform molecules is not the aim of this minireview. The new development that promotes the utilization of biomass is given in this minireview.

## 2. Structural Composition of Lignocellulosic Biomass

As the only renewable carbon-based resource, biomass was recognized as both energy and raw materials; more than 120 billion tons every year are produced on a global scale, equalling 2.2 × 10^21^ J of energy [5]. Biomass derived from various raw sources consists of 30–60% cellulose, 20–40% hemicellulose, and 15–25% lignin [6]. These resources contain sugar cane bagasse, switchgrass, rice straw, corn stover, corn cobs, nutshells, grasses, wheat straw, and so on.

Lignocellulose is composed of three main components: cellulose, hemicellulose, and lignin; in addition, small amounts of pectins and proteins are present in it. The existence of cellulose and hemicellulose in lignocellulosic biomass draws much attention; unfortunately, lignin usually encloses in or protects them, resulting in the formation of crosslinkages [7] and the recalcitrance of lignocellulosic biomass. As cellulose consists of glucose monomers linked with β-1,4-glycosidic bonds and makes up the largest fraction of lignocellulose [8], the hydrolysis of which can release glucose. The second predominant component is hemicellulose, which consists of C5 sugars such as glucose, xylose, mannose, galactose, and arabinose [9], the amorphous polymer of which is xylan [10]. Unlike cellulose, the structure of hemicellulose is amorphous [11]. Hence, it is the least thermo-chemically stable constituent of lignocellulose [7] and is not isolated, different from cellulose. Cellulose and hemicellulose are the most frequently introduced biomass raw material to produce furan derivatives [12,13]. Lignin is a complicated, three-dimensional amorphous, and robust biopolymer [14], consisting of phenylpropane units, derived from sinapyl alcohol, coniferyl alcohol, and *p*-coumaric alcohol [15]. The structure of lignin is complicated and can differ depending on the species [16,17], temperature [18,19], and environmental history of biomass resources [20,21] (Table 2).

## 3. Efficient Solvent Systems for Lignocellulosic Biomass Conversion

For non-edible feedstock, lignocellulosic biomass as the primary resource has drawn much attention to produce biofuels, which is extremely significant due to the potential to reduce the environmental and geopolitical impacts caused by fossil fuels. Among the biofuels, various furanic derivatives, such as HMF and 2,5-dimethylfuran (DMF), as important platform molecular appear in the “TOP 10 + 4” depicted by the US Department of Energy. These TOP molecules can act as feedstock for the synthesis of valorization products, containing fuel additives, polymer monomers, solvents, and other chemicals.

### 3.1. Ionic Liquid System

As a type of significant and green solvent, ILs have drawn increasing attention for the conversion of lignocellulosic biomass to chemicals and biofuels [22,23,24,25] because of the characteristics of designability and recyclability [26,27,28]. Large organic cations and inorganic or organic anions make up the low melting point, high thermal and chemical stability, and easy separation liquid phase [29]. Application of ILs has devoted contributions to the conversion of lignocellulosic biomass [30,31,32,33,34,35,36] since Rogers’ group announced dissolving cellulose in [C4mim]Cl in 2002 [30], which deeply vitalized their introductions for marine biomass. Among reaction processes, ILs enhanced catalytic activities [37] by destroying the intramolecular/intermolecular hydrogen bonds of cellulose and hemicellulose.

Nargotra et al. [38] pretreated sunflower stalks with different concentrations (10–25%, *w*/*w*) of 1-butyl-3-methylimidazolium chloride (BMIMCl) and reported a maximum of 25% *w*/*w*, providing a higher total reducing sugars. Furthermore, at higher NaOH concentrations (>0.5%, *w*/*v*), sugar release was decreased. Nguyen et al. [39] developed the two stage of ammonia with 1-ethyl-3-methylimidazolium acetate ([EMIM] Ac), providing higher sugar yield and increased conversion of glucose with 74% cellulose recovery and 78% glucose conversion. Different acetate and chloride anion groups of ILs revealed various effects on the cell wall of Eucalyptus, emphasizing the importance of anions over the cations in the degradation process [40]. Sriariyanun et al. [41] explored combined effects of a screw press with different ILs ([BMIM]Cl, 1-ethyl-3-methylimidazolium chloride ([EMIM]Cl), and [EMIM] Ac, 1-allyl-3-methylimidazolium chloride ([AMIM]Cl). [AMIM]Cl afforded 76% reduction sugar with enhanced saccharification, reaction time saving, and more efficiency. Zhang et al. [42] reported the application of [BMIM]Cl with various metal chlorides (e.g., CrCl_3_, CuCl_2_⋅2H_2_O, CrCl_3_/LiCl, FeCl_3_⋅6H_2_O, LiCl, CuCl, and AlCl_3_) for the formation of furfural. AlCl_3_ exhibited the highest furfural yield (77%) when xylan was used as the feedstock, and the Al^3+^ ionization potential was a determining factor for the excellent furfural yield. It was proposed that AlCl_3_ reacted with [BMIM]Cl to form [AlCl_n_]^(n^
**^−^**
^3)−^ complexes which weakened glycosidic oxygen atoms to hydrolyze xylan to produce xylose and also isomerize xylose as an enolate structure to dehydrated furfural. Under optimal conditions, AlCl_3_ combined [BMIM]Cl to yield furfural derived from corncob, pinewood, and grass, with 19.1%, 33.6% and 31.4%, respectively.

To take advantage of the dissolving abilities of ILs and their strong acidic nature, Brønsted acidic ILs (BAIL) such as [C_3_SO_3_HMIM][HSO_4_], [C_3_SO_3_HMIM][PTS], and [C_3_SO_3_HMIM][Cl] were developed to act as solvent and catalyst. Particularly, [C_3_SO_3_HMIM][HSO_4_] facilitated the hydrolysis of hemicellulose, showing higher sugar yields and furfural yields of up to 85% from beechwood because of the ion–dipole interactions between [C_3_SO_3_HMIM][HSO_4_] and hemicellulose and the higher acidity of BAIL.

Furthermore, [C_3_SO_3_HMIM][HSO_4_] was stable in the reaction conditions and recycled for four cycles with minimal loss of inactivity [43]. Wheat straw hemicellulose was also selectively and effectively hydrolyzed into xylose and arabinose in [EMIM][HSO_4_]. The yield and the recovery of pentoses from the reaction liquor afforded 80.5 and 88.6%, respectively. Meanwhile, ILs could be reused with a high yield of 92.6 wt% and recycled with negligible selectivity of hemicellulose hydrolysis [44]. Alam et al. [45] revealed the preparation of sulfonic acid functionalized BAILs using multiple cations and anions for the conversion of wood ear mushroom to HMF, demonstrating the following catalytic performance order: [BBIM-SO_3_][NTf_2_] > [BBIM-SO_3_][OTf] ≈ [DMA][CH_3_SO_3_] > [NMP][CH_3_SO_3_]. The obtained trend can be relevant to their proton donating ability, which is correlated with the DFT calculated values of the deprotonation energy of the BAILs, and the high activity of [NTf_2_]^−^ anion is proposed for the strong electron-withdrawing properties.

Chen et al. [46] provided an effective strategy to produce methyl levulinate (ML) from aquatic microalgae duckweed. Results confirm that the infrared structure has a significant effect on its acid strength, which eventually determined the efficiency of ML formation. With the optimized catalyst [C_3_H_6_SO_3_HPy]HSO_4_, the conversion rate of duckweed was 88.0%, and the yield of ML was high, up to 73.7% at 170 °C for 5 h. Beyond that, the solvent changed from methanol to water, and the process efficiency decreased from 81.8% to 53.7%, which further revealed that the significantly enhanced performance of the solvent was devoted to the high efficiency of the in situ esterification of the LA. Wang et al. [47] investigated the Lewis acidic ILs (LAIL) such as [BMIM]Cl–AlCl_3_ as a catalyst for furfural formation in an H_2_O–GVL biphasic system. The exploration of the dissolving capabilities of ILs and the furfural yields demonstrated that corncob showed the best furfural yield with 47.96% due to the high hemicellulose content (31.79%). Results confirm that [BMIM]Cl–AlCl_3_ would form [AlCl_4_]^−^ that affected the normal hydride shift in the xylose–xylulose isomerization process for the production of furfural.

Naz et al. [48] developed various Bronsted acidic ILs (BAILs): Bronsted Lewis acidic ILs (BLAILs), Lewis acidic Ils (LAILs), as well as organic electrolyte solutions (OES) for HMF synthesis derived from monosaccharides, polysaccharides, and even lignocellulosic composites. [C_4_SO_3_HC1Py]AlCl_4_ is the most efficient catalyst in converting fructose and glucose to HMF but is inefficient for cellulose and wheat husk biomass. The reason can be ascribed to the low isomerization and saccharification power of SO_3_H groups united with the side chain of cation, the removal of which provides 54% cellulosic conversion and 35% of wheat husk. A series of pyridinium-based ILs were investigated for efficient conversion of fructose, glucose, galactose, cellulose, and real biomass. A total of 99% HMF was obtained with [C_6_Py]Cl without any other catalyst from fructose in 2 h at 80 °C, catalyzed by best isomerization and saccharification catalyst CrCl_3_ along with [C_4_C_1_Py]Cl affording 82 and >70% yield from glucose and cellulose at 100 and 120 °C, respectively. With the same catalytic system, 20–35% HMF yield from agricultural residues, as well as dry fruit shells, were provided [49].

Despite being promising for lignocellulose biomass conversion, some disadvantages still limit the practical applications of ILs. Generally, the widespread use of choline-based ILs is because of the advantages, namely biocompatible, renewable, lower costs, and easily available. However, the acetates also exist unstably and degrade with time and temperature and volatile defects. The HSO_4_^−^-based protic ILs are well-known for their economical cost and are limited by their corrosiveness and hygroscopicity. Among the acidic ILs, the bisulphate-based ones are the most active and the small-sized cations showed better performance over the large-sized ones. Meanwhile, the synthesized ILs with SO_3_H as a functional group on the cation have the advantage of avoiding the drying and purification steps, which happened similarly by using the COOH functional group [50].

### 3.2. Deep Eutectic Solvent System

Compared with ILs, DESs have assumed favorable merits, such as high thermal stability, non-flammability, tunable physicochemical properties, availability of starting materials, superior biodegradability, easy synthesis, and good recyclability [51,52]. DESs were first introduced by Abbott and co-workers and are eutectic mixtures formed by hydrogen bond donors (HBDs) and hydrogen bond acceptors (HBAs) [53]. DESs have been applied in lignocellulosic biomass catalytic conversion, biomass pretreatment, carbohydrate product conversion, and lignin extraction and upgrading [53,54,55,56,57,58,59,60]. DESs are mainly composed of ChCl and various HBDs, including carboxylic acids, alcohols, and amides, and are the most commonly employed in biomass processing [51,61]. Many elements influence the performance of DESs on biomass catalytic conversion, such as reaction time, reaction temperature, liquid/solid ratio, DESs type, and HBD/HBA ratio [62], among which, the type and percentage of functional groups in DESs mixture are of great importance.

Zhang and Yu [63] explored the choline chloride-citric acid (ChCl-citric acid) DES as the reaction medium and Brønsted acidic catalyst to prepare furfural. Comparing co-catalysts including AlCl_3_⋅6H_2_O, FeCl_3_⋅6H_2_O, and CrCl_3_⋅6H_2_O with no metal chloride, 51.3% furfural yield was produced from xylan with the introduction of AlCl_3_⋅6H_2_O under optimal reaction conditions. The addition of metal chlorides enhancing furfural yields is of great importance, which ascribes to the production of Lewis and Brønsted acid reaction medium. DESs facilitated the hydrolysis of xylan and Lewis acids, increasing xylose isomerization to xylulose, whereas Brønsted acid catalyzed the dehydration of xylulose to furfural. AlCl_3_⋅6H_2_O afforded the highest furfural yield because of the higher ionization potential of Al^3+^, which was further confirmed when no furfural was detected in the water in place of DESs under low operating temperature (100 °C). Various DESs (ChCl-urea, ChCl-oxalic acid, and ChCl-betaine) were investigated for the synthesis of furfural and HMF with lignocellulosic waste as raw materials. ChCl-oxalic acid with TiO_2_ afforded 9.96 wt% furfural yields compared with ChCl-urea-combined hydrotalcite (5.25 wt%) and ChCl-betaine with Na_2_MoO_4_ (8.63 wt%) for catalytic performance studies using cellulose as feedstock. The addition of water enhanced the HMF and furfural yields, which was attributed to the catalyst solubility increasing in the reaction system, while excess water (>2 mL) led to low yields for the production of degradation products. Kinetic investigation found that furfural concentrations of 0.876 g/L, 0.747 g/L, 0.470 g/L, and 0.336 g/L were derived from macauba shells, pine nutshell, carnauba leaves, and macauba pulp under optimum 140 °C, respectively. Increasing the accessibility of chemical reactions by introducing ultrasonic irradiation to break down lignocellulosic biomass explained the growth of furfural and HMF yields (14.99 wt% to 36.59 wt% for macauba pulp [64]. Rice straw was conducted by a lactic acid/ChCl–water system with HBA and HBD, demonstrating efficient breakage for the linkage between hemicellulose and lignin profit from their synergistic effect. Highs of up to 70% xylan were removed and fractionated into the liquid stream consisting of xylose, furfural, and humins. DESs could be recycled with a high yield (69%) after five times at 90 °C [65].

Experiment results demonstrate that HBDs with carboxyl groups (e.g., formic acid, oxalic acid, and lactic acid) showed more preferable performance in lignocellulosic biomass pretreatment and delignification than those containing other groups (e.g., hydroxyl groups), regardless of the HBA utilized. However, the molar ratio of HBD to HBA is high to a certain extent; excellent delignification efficiency is mainly enhanced by the provided more active protons by carboxylic acids, which facilitate the proton-catalyzed cleavage of chemical linkages in the lignocellulose.

Several IL-based DESs were investigated by introducing thiourea (TU), N-methylthiourea (NMTU), glycerol (Gly), and ethylene glycol (EG) as the HBDs and four ILs, i.e., [BMIM]Cl, [EMIM]Cl, [HMIM]Cl, and [AMIM]Cl as the HBAs (Figure 1). It was found that the synthesized DESs showed good performance on the dissolution of xylan, especially for [AMIM]Cl-EG (2:1), with the best solubility of 40.4 wt% at 70 °C. To our delight, the prepared DESs emerge with great potential in the transformation of lignocellulose, and the development of DESs is a promising approach to enhancing the dissolution and conversion of biomass [66].

The hydrolysis of hemicellulose can convert to xylose or arabinose by acidic DESs, which is further dehydrated to prepare furfural by releasing three water molecules. It is critical to optimize and mediate the acidic strength of the acid catalyst/solvent for the hydrolysis of hemicellulose; the higher the pH grew closer to 1, the larger the productivity of furfural [67]. When the 1:1 molar ratio of Brønsted acidic DESs and natural acidic DESs was observed, catalyzation by choline chloride/p-toluene sulfonic acid (ChCl:p-TSA = 1:1) showed the best yield of furfural (85.4%) under pH = 1 at 120 °C for 1.5 h. Increasing the pH from 1 to 3, 51.4% furfural yield was achieved. The effect of acidity on the furfural yield was also similarly demonstrated by Cornelius et al. using choline chloride-dicarboxylate-based low-eutectic solvents.

A one-pot, two-step approach was developed for levulinic acid production derived from rice straw (Figure 2). The investigation of the effect of carboxylic-acid-based DESs (choline chloride-acetic acid, choline chloride-oxalic acid, choline chloride-malonic acid, and choline chloride-succinic acid) confirmed that choline chloride-oxalic acid DESs observing the highest levulinic acid yield (52%) by HCl at mild condition (120 °C at 10 h). The synthesis of DESs provided a renewable, low toxicity, and cheap-ingredients approach for highly efficient, reduced costs in levulinic acid preparation. This greener process has excellent potential for application to other lignocellulosic biomasses [68].

### 3.3. Biphasic System

Compared with a single-phase reaction medium, biphasic systems are composed of a reaction phase and extraction phase, which can shift the prepared furan derivatives from the reaction phase to the extraction phase immediately, inhibiting the degradation of production and then enhancing the target chemical’s formation efficiency. Generally, biphasic systems consisting of water and organic solvents have widely acted as the reaction medium for the synthesis of significant platform chemicals such as HMF. In this aspect, methyl isobutyl ketone (MIBK), butanol, and Tetrahydrofuran (THF) are the most utilized organic solvents because of their suitable partition coefficients and salts, NaCl especially is introduced to facilitate the transfer of target chemicals from the water phase to the organic phase via the salting-out effect.

Zhang et al. [69] published a furfural synthesis introducing an H_2_O–Toluene biphasic system with corncob as feedstock catalyzed by CrCl_3_⋅6H_2_O and NaCl. As the stronger Lewis acidity of CrCl_3_.6H_2_O compared with AlCl_3_⋅6H_2_O, CuCl_2_⋅6H_2_O, FeCl_3_, CoCl_2_⋅6H_2_O, and ZnCl_2_, a furfural yield of 14.31% was obtained. NaCl enhanced the partition coefficient of furfural in the organic phase, which limits furfural degradation, thereby improving yield. With the co-catalyst NaCl, the furfural yield observed grew to 23.88%. Similarly, Yang et al. [70] prepared furfural from various biomass including corn stover, pinewood, and switchgrass in an H_2_O-THF biphasic reaction medium catalyzed by the Lewis acid catalyst (AlCl_3_⋅6H_2_O) and co-catalyst NaCl. Furfural yields were achieved from 38% to 64%, suggesting the isomerization of xylose to xylulose and the subsequent dehydration to give the target molecule. The combination of NaCl, water, and organic solvent afforded excellent furfural yields derived from pubescent. The first process was the decomposition of hemicellulose, lignin, and cellulose in H_2_O-GVL. The water exhibited better dissolution for hemicellulose (96.5%) compared with GVL (11.5%), but no furfural was observed. Without GVL, only a 2.3% yield of furfural was achieved in comparison with 46.6% when in the presence of GVL. As no catalyst was introduced, the production of furfural through dehydration reaction was catalyzed by the acidic condition generated by the dissolution of biomass providing organic acids (e.g., formic acid and acetic acid). Consequently, dissolved hemicellulose oligomers were conducted to synthesize furfural with the appendage of THF and NaCl, with the existed GVL gained from the former step [71].

Owing to the low toxicity, biodegradability, and high partition coefficient, 2-methyltetrahydrofuran (MeTHF) produced from biomass-derived furfural or levulinic acid is considered a promising alternative to THF. Meanwhile, ethyl acetate, n-propyl acetate, and isopropyl acetate were also the preferred solvents based on their performance, environmental health, and safety impacts integrally.

Except for the generally reported organic phase, AlCl_3_-catalyzed DESs/MIBK biphasic systems on lignocellulosic biomass for furfural and glucose were invested because of their lignocelluloses-dissolving ability and furfural production improvement of DES. The renewable DESs (choline chloride-oxalic acid) served as both pretreatment solvent and Brønsted acid catalyst. At the optimum condition (at 140 °C for 90 min), the best yield of furfural was 70.3%, and the saccharification yield was high, up to 80.8%. The AlCl_3_-catalyzed DESs/MIBK biphasic system pretreatment could realize the high-value utilization of lignocellulose [72].

## 4. Conclusions

This work summarized and discussed the latest developments on lignocellulosic biomass conversion to various molecules, especially furan derivatives in different reaction systems. Most systems are effective for biomass catalytic conversion; however, the investigations are still in their infancy and many issues hinder the further exploration and even development of the industry, as follows:(1)To discover an optimized and cost-effective system for lignocellulosic biomass conversion;(2)The research on DESs recycling and stability needs to be studied in-depth;(3)The thorough exploration of solvent effect mechanisms;(4)The commercial synthesis of furan derivatives from biomass on a large scale requires further innovation;(5)Assistive technologies such as supercritical CO_2_, microwave, and ultrasound have been introduced into the various reported solvent system; the synergistic effect should be realized effectively.

## Figures and Tables

**Figure 1 polymers-14-03671-f001:**
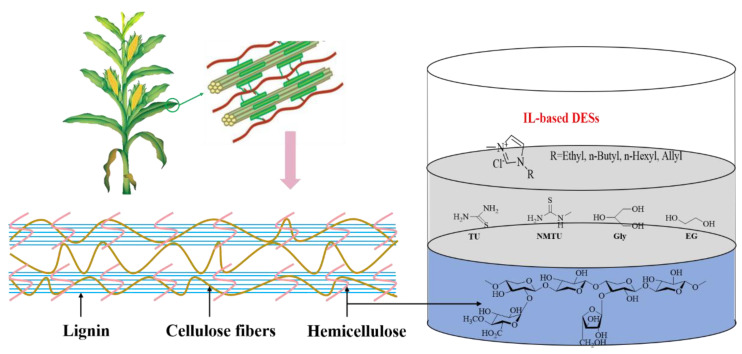
IL-based DESs were developed for highly efficient dissolution of xylan.

**Figure 2 polymers-14-03671-f002:**
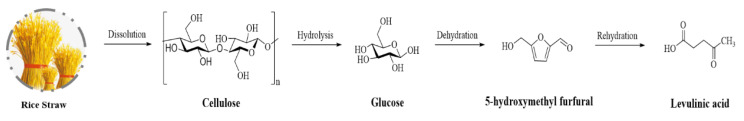
Reaction pathway of levulinic acid synthesis from lignocellulose.

**Table 1 polymers-14-03671-t001:** Single, two-stage or multi-stage approaches.

	Advantages	Disadvantages
Single-stage	Easy to handle	Longer residence time, higher energy consumption, low yield, corrosion, failure to industrialize
Two-stage	Fast reaction rate, moderate yield	Partial not green, tedious industrialize
Multi-stage	Reduced sugar degradation, improved sugar yields, require less energy	Hard to manipulate

**Table 2 polymers-14-03671-t002:** Components of lignocellulosic biomass and the derived furanic derivatives and others production.

Component	Monomer	Reaction	Furanic Derivatives and Others
Cellulose	Glucose	Fermentation	H_2_, ethanol, lactic acid, succinic acid, acetic acid
Hemicellulose	Glucose, xylose, mannose, galactose, and arabinose	Hydrolysis	Reducing sugars
		Hydrogenation	Xylitol, sorbitol
Lignin	Coniferyl, p-coumaryl, sinapyl alcohol	Isomerization	Fructose, xylulose
		Dehydration	5-hydroxymethylfurfural (HMF), furfural, 2,5-bishydroxymethylfuran, γ-valerolactone (GVL)

## Data Availability

Not applicable.

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
