# Peer review of "Efficient Reaction Systems for Lignocellulosic Biomass Conversion to Furan Derivatives: A Minireview"

_polymers, 2022, doi:10.3390/polym14173671_

Round 1
Reviewer 1 Report
This review summarizes the process of obtaining high value-added furanic derivatives from lignocellulosic biomass, a renewable resource. Particular attention was paid to ionic liquids, DES, and biphasic systems. It is highly commendable that the author not only lists the several research results but also discusses the challenges and future prospects. On the other hand, although it is stated that this is a mini-review, it seems to lack comprehensiveness, especially with regard to recent research results. In addition, although there are not many, it contains obvious errors. Appropriate corrections are required before publication.
Major comments
1. Several reviews have already been published on studies of obtaining furan-based compounds from lignocellulose, e.g., “Recent advances in biotransformation of 5-Hydroxymethylfurfural: challenges and future aspects”, JOURNAL OF CHEMICAL TECHNOLOGY AND BIOTECHNOLOGY 97 (2) , pp.409-419.
Please explain what makes this review different from previous review articles, i.e., the significance of this compilation and its impact on the scientific community.
2. Although the abstract clearly states "recent trend," I have the impression that the number of citations for very recent papers after 2020 is small. For example, when searching for papers with the keywords “HMF, ionic liquids, lignocellulose”, many papers published after 2020 were found. Some of them are listed below, but this is not all. Please research again and correctly add more recent papers, or indicate any reasons why you did not include these papers.
Examples:
Biomass-Derived Ionic Liquids Based on a 5-HMF Platform Chemical: Synthesis, Characterization, Biological Activity, and Tunable Interactions at the Molecular Level
Seitkalieva, MM; Vavina, AV; (...); Ananikov, VP
Mar 8 2021 | Feb 2021 (Early Access) | ACS SUSTAINABLE CHEMISTRY & ENGINEERING 9 (9) , pp.3552-3570
Effect of molecular structure of cation and anions of ionic liquids and co-solvents on selectivity of 5-hydroxymethylfurfural from sugars, cellulose and real biomass
Naz, S; Uroos, M and Muhammad, N
Jul 15 2021 | May 2021 (Early Access) | JOURNAL OF MOLECULAR LIQUIDS 334
One-pot production of 5-hydroxymethylfurfural and simultaneous lignin recovery from non-food lignocellulosic wastes using cost-effective ionic liquids
Naz, S; Uroos, M and Muhammad, N
Jan 2022 (Early Access) | BIOMASS CONVERSION AND BIOREFINERY
Ionic Liquids Mediated One-Pot Synthesis of Second Generation 5-Ethoxymethylfurfural (5-EMF); A Potent Biofuel Candidate
Naz, S and Uroos, M
May 13 2022 | CHEMISTRYSELECT 7 (18)
Minor comments
3. L81: I believe it is the melting point, not the boiling point. With a few exceptions, most ionic liquids do not have boiling points.
4. L85: I do not recall any mention of application to marine biomass in the cited reference 29. For now, there have been only a few reports on the use of ionic liquids for the treatment of marine biomass, and it has not been a major research topic.
Author Response
Thank you very much for providing further suggestions on our manuscript entitled “Efficient Reaction Systems for Lignocellulosic Biomass Conversion to Furan Derivatives: a minireview” submitted to Polymers. The paper has been modified by carefully following the Reviewers’ Comments. All the queries raised by the Reviewers have been addressed point-wise and remedial measures adopted accordingly.
Other English and related technical changes are indicated in blue and are self-explanatory.

Reviewer 2 Report
The presented study shows a mini-review on lignocellulosic biomass converted to furanes. The study seems novel and useful. An overview of different techniques and their pros and cons is presented. The future research directions are indicated.
Some technical remarks are shown below.
1. Please, change the last sentence in the Abstract section to follow the logical path.
2. Please pay attention that there is no more than one space between the words and sentences (line 29), that every sentence finishes with a dot (line 33), etc. Change „minireview“ to „mini-review“. The phrase would be good to be included in the title. Change „Furanic Derivatives” to “Furan Derivatives” in the complete manuscript.
3. Please explain in more detail the sentence in lines 82-85.
4. Also, explain which production of sugars and what type of its decrease is discussed in lines 88-90.
5. For better readability, avoid using the same words in one sentence more than once (For example: „Furfural yields achieved 248 from 38% to 64%, suggesting the isomerization of xylose to xylulose and the subsequent 249 dehydration for furfural.”). Please check the whole text.
6. Define GVL at the first mention.
7. Do not mention abbreviations if the term is mentioned only once in the text, like EtOAc. Check if there are others.
8. Improve the conclusion section. Change “Most systems are effective for biomass catalytic conversion; however, the investigations are still in their infancy and many issues hinder the further exploration and even development of the industry. As follows:” to “Most systems are effective for biomass catalytic conversion; however, the investigations are still in their infancy and many issues hinder the further exploration and even development of the industry, as follows:”
-Please explain in detail all the 6 points made, to follow the previous claim. What are the constraints of further development of industrial-scale production? In which way the in-depth study should be carried out? In which direction should solvents be further studied? You need to be more specific about all the conclusions because these are the most important part of the mini-review. Also, include further conclusions both in the abstract and introduction sections.
Author Response

(The authors gave the same response as above.)

Reviewer 3 Report
Abstract
Need to improve by include the significant of this review.
Introduction
Line 27-29: Please mention reference.
What is furanic derivatives. ?
“Utilization of lignocelluloses”.. is it drawback for line 31-33 ?. If yes, then start with However,….. in order to show the correlations ?
Please compare “single, two-stage or multi-stage approaches” in table. Please include the energy consumption data.
Please details out the problem statement regarding to conventional reaction system that leads for Efficient Reaction System that based on the title of paper.
Please recap the objective of this paper in the last para of introduction.
Section 2
Please major revise this section. It is not strong that discuss about the Structural composition of lignocellulosic biomass. I need table of comparison, related figures, and related it with furanic derivatives.
Section 3
Please start this section with why need to “lignocellulosic biomass conversion”
Please conclude each method in table. Include also the advantages and disadvantages.
Please separate line 108-139 into 2 or 3 para.
Please add figure that describe Ionic liquid system. Same goes to other method ?
Major comments:
Please add one more section that highlight the furanic derivatives form biomass in applications (section 4)
Other than (1) ionic liquid; (2) deep eutectic solvent; (3) biphasic system; how about physical treatment, mechanical, physicochemical and biological treatment ?
Conclusions:
I don’t think the sentence is correct à “Most systems are effective for biomass catalytic conversion; however, the investigations are still in their infancy and many issues hinder the further exploration and even development of the industry.”, since you may found a of literature and study that discuss the methods. Authors may conclude the best method, disadvantages and the applications of that method. Then continue with the stated recommendations.
Author Response

(The authors gave the same response as above.)

Round 2
Reviewer 3 Report
After a second revision, I dont think this review Q1 paper is strong enough to be accepted. Looking at specifically at section 3 it just a summary of literatures method.
While for Section 2 is too common. Nothing is new and correlate with Furanic term.
Round 3
Reviewer 3 Report
The title of this paper is "mini review", and focusing on furanic derivatives. Then this paper is accepted.
Please check the usage of capital and small letter.
